# Structural Response of Bonded Joints between FRP Composite Strips and Steel Plates

**DOI:** 10.3390/ma14216722

**Published:** 2021-11-08

**Authors:** Vlad Lupășteanu, Dragoș Ungureanu, Nicolae Țăranu, Dorina Nicolina Isopescu, Radu Lupășteanu, Petru Mihai

**Affiliations:** 1Faculty of Civil Engineering and Building Services, “Gheorghe Asachi” Technical University of Iaşi, 43 Mangeron Blvd., 700050 Iaşi, Romania; vlad.lupasteanu@academic.tuiasi.ro (V.L.); nicolae.taranu@academic.tuiasi.ro (N.Ț.); dorina-nicolina.isopescu@academic.tuiasi.ro (D.N.I.); radu.lupasteanu@academic.tuiasi.ro (R.L.); petru.mihai@academic.tuiasi.ro (P.M.); 2The Academy of Romanian Scientists, 54 Splaiul Independentei, Sector 5, 050094 Bucuresti, Romania

**Keywords:** polymer–matrix composites, bonded joints, epoxy adhesives, finite element analysis (FEA)

## Abstract

This paper presents the outcomes of an experimental and numerical study performed on epoxy-bonded single lap joints (SLJs) between carbon fiber-reinforced polymer (CFRP) composite strips and steel elements. For the experimental program, 34 specimens were prepared by varying the type of the composite strip and the type of adhesives and their thicknesses; all specimens were loaded in axial tension up to failure. The specific failure mechanisms were identified and commented on the basis of the performed tests, and the load–displacement curves were plotted. Additionally, the strain distributions along the bond lengths at different load stages, the shear stress–displacements (slip) variations and the stress–strain distributions for the CFRP strips were plotted and investigated. The numerical simulations, based on 3D finite element method (FEM) analysis, provided consistent results, in good agreement with the experimental ones for all parameters that were investigated and discussed in this paper.

## 1. Introduction

Over the past decades, composite materials have been widely used to strengthen existing structural elements made of traditional materials [1,2,3]. The use of fiber-reinforced polymer (FRP) composite products (laminates, plates, fabrics and sheets) to repair and strengthen reinforced concrete and masonry structures is well established [4,5,6]. On the other hand, the application of FRP materials to steel structures remains understudied since traditional methods of repairing and strengthening are still highly used. These methods consist in either adding supplementary metallic elements to increase the cross-sectional dimensions or embedding the existing steel elements into concrete sections [7]. However, the traditional strengthening methods of steel structures have two main disadvantages. The first one refers to the fact that the added elements have considerable weight, and the installing technologies may further affect the damaged element (the case of the residual stresses that develop due to the welding process). The second disadvantage refers to the time needed to carry out the intervention works, which is often quite long, thus affecting the serviceability of the construction.

Compared to the traditional methods used for strengthening, the techniques based on FRP composite materials have remarkable advantages that arise from their superior mechanical characteristics and corrosion resistance properties [7,8,9,10]. The structural efficiency of these methods was analyzed by various research teams by conducting experimental programs based on numerical analysis methods and by adjusting and applying the existent theoretical models specific to the hybrid consolidation solutions [11,12,13,14]. However, the progress in this research area has been achieved mainly through the relatively recent development of FRP composite products that are structurally compatible with steel elements.

Most of the FRP composite products used to repair and strengthen steel elements are attached to their surfaces by an adhesive layer. The main function of the adhesives is to ensure the contact between the steel element and the FRP one, thus guaranteeing an efficient transfer of the stresses. Therefore, the interface behavior of the FRP composite products bonded to steel surfaces is a key aspect in assuring an efficient strengthening solution for structural or non-structural steel elements [15].

This paper describes an experimental and numerical study performed at the Faculty of Civil Engineering and Building Services of Iasi. The study included 34 single lap joints (SLJs) specimens, each of them being composed of a carbon fiber-reinforced polymer (CFRP) strip-bonded to a steel element. The single-lap joints are among the most common types of bonded connections, being utilized in various engineering applications [16,17]. Each specimen was instrumented so that the load, the slip between the adherents and the axial strains of the un-bounded part of the CFRP strip could be monitored during the tension tests. All specimens were tested up to failure using the load control procedure. The same typologies of bonded connections were numerically studied using the finite element method (FEM) analysis. A good agreement was obtained between the results provided by the experimental program and the ones obtained through numerical analysis.

The main objectives of the research program consist in studying, based on simplified models, the suitable configuration of bonded joints between common CFRP strips and steel elements, with respect to the type and the thickness of the adhesive. The bonding behavior is one of the key parameters that influence the structural response of steel elements which are strengthened with CFRP products.

## 2. Experimental Program

### 2.1. Specimens Description

The main aim of the carried out experimental program consists in the investigation of the specific characteristics of the interface behavior between steel elements and CFRP strips bonded with adhesives. During the design and preparation stages of the specimens, the following parameters were taken as variables: the axial stiffness of the CFRP composite strips, the moduli of elasticity and the thicknesses of the adhesives and the surface preparation method for the steel substrates. Thus, two types of CFRP composite strips (Sika Carbodur S512 and Sika Carbodur M514) [18,19] were bonded to either prepared or unprepared steel surfaces using two types of adhesives (Sikadur 30 and Sikadur 330) [20,21], having three different thicknesses (1 mm, 2 mm, 3 mm). For each combination, at least two identical specimens were prepared. For all specimens, the bond length was 350 mm, being higher than the effective one, calculated using the analytical model given by [22,23].

### 2.2. Materials Properties

The steel plates used to assemble the specimens were cut from a 10 mm thick plate using a high-speed circular metal sawing machine. After the cutting process, the edges of the steel plates were rectified and polished in order to obtain the design dimensions of the specimens. The geometrical features and the mechanical properties of the steel plates are given in Table 1.

The CFRP strips used to assemble the specimens belong to the Sika’s Carbodur composite products range. The selection of the two types, S512 and M514, was made by taking into account one of the main objectives of the experimental program, which is to determine the influence of the axial stiffness of the CFRP strip on the shear structural response of the bonded specimens. Thus, the two types of CFRP strips have similar tensile strength but different moduli of elasticity, namely 165 GPa for S512 and 210 GPa for M514 [18,19]. The most relevant properties of the CFRP composite strips, as provided by the manufacturer, are presented in Table 2.

The adhesives used for bonding the CFRP composite strips to the surface of the steel plates, Sikadur 30 and Sikadur 330, are provided by the same manufacturer and are adequate for this kind of hybrid bonding system. Both products are thixotropic, consisting of a combination of epoxy resins, additives and hardening agents. In addition, from a mechanical point of view, both adhesives have linear-elastic behavior. As it can be observed in Table 3, the tensile strengths and the ultimate strains of the adhesives have similar values, while the modulus of elasticity of Sikadur 30 is almost three times higher than that of Sikadur 330 [20,21]. The values presented in Table 3 are given by the producer, being valid for specific conditions of preparing and curing the adhesives, which were carefully ensured during the experimental program.

### 2.3. Experimental Set-Up

After all the necessary materials for assembling the specimens were shaped to their nominal dimensions, the first step consisting in preparing the steel surface was performed. According to the Mechanical Theory, the degree of adhesion that may be obtained for an adhesively bonded joint is directly linked to the roughness of the substrates [25,26]. Thus, 30 of the 34 steel plates were sandblasted to increase the specific roughness of the surface that comes into contact with the adhesive [27]. The remaining four steel plates were not sandblasted so as to determine the effect of this surface treatment method. The sandblasting process was performed in a special room with a controlled environment using GL-80 steel grits made by W Abrasive that was shot onto the surface of the steel plates from a distance of 5 cm and at an inclination angle of 45° (Figure 1, Table 4) [28]. After sandblasting, the steel plates were vacuumed and wrapped in a plastic foil to prevent future contamination.

The bonding of the CFRP strips was performed within a maximum of 24 h of sandblasting the steel plates. For each specimen, the CFRP strips were cleaned with a special solvent (Sika ColmaCleaner) [29] to remove all the protective coatings that could affect the quality of the bonds. The adhesive was then prepared by mixing the components, as indicated in their technical data sheets [20,21].

One of the most applied techniques of bonding composite products on different substrates recommends that the adhesive should be applied initially to the composite strip and then pressed on the base element [30,31]. However, this technique is particularly suitable for site applications where the technological phases have to be carried out within a limited time. In addition, when the bond length is relatively small (close to the effective one) and the thickness of the adhesive layer is fixed, it is difficult to determine the amount of adhesive that should be applied to the composite strip. In this case, two factors with a negative impact on the mechanical performances of the bond may occur. The first refers to a bond area that is insufficiently covered with adhesive, which generally results in a lower thickness than the one desired. The second consists in applying the excess adhesive which slumps during the pressing of the composite element on the base member. On the other hand, when adhesive joints are assembled under laboratory conditions and their geometrical tolerances are minimal, it is necessary to use specific control methods to ensure that the designed geometrical characteristics are obtained. For this purpose, in order to obtain a constant thickness of the adhesive layer, chrome-steel spherical spacers of 1, 2 or 3 mm were fixed on the surface of the steel plate, depending on the type of the specimen. These spacers do not affect the structural behavior of the joint since the voids that may be formed near them collapse to a negligible volume when the specimens are fixed with clamps. Additionally, the spherical steel spacers (and/or the calibrated wires) are recommended by practice guides as some of the most suitable methods to control the adhesive thickness [32]. A special rig was designed (Figure 2b) to control the quantity and the homogeneity of the adhesive that is applied on the surface of the CFRP strip. Figure 2a presents the laboratory setup of the rig. After the assembling stage had ended, the specimens were fixed with clamps and stored in constant temperature and humidity conditions for 14 days until the curing process was completed.

Each specimen was marked with a code (i.e., S1 – S512-330-1-II), indicating the type of the CFRP composite strip, the type of adhesive and its thickness. The last term stands for identifying specimens with identical configurations. The general layout of the specimens is presented in Figure 3. In Table 5, the first five columns show the configurations of the specimens, while the last two columns list their corresponding failure loads (P_ult_) and failure mechanisms recorded during the experimental tests.

### 2.4. Specimens Instrumentation and Loading Conditions

The parameters that were monitored during the shear pull-of tests are the applied tensile force, the relative displacements between the CFRP strip and the steel plate and the variation of the strains along the composite element. The shearing effect in the adhesive layer was obtained by loading all specimens in tension at their free ends in a Zwick/Roell 100 kN hydraulic test machine (ZwickRoell, Ulm, Germany) (Figure 4), located in the Composite Materials Laboratory of the Faculty of Civil Engineering and Building Services, Iași.

All tests were performed under force-control by imposing a loading rate of 5 kN/min. The ultimate forces were recorded by the testing machine connected to the acquisition system. The displacement between the CFRP strip and the steel plate was recorded using a linear variation displacement transducer (LVDT) mounted on the two elements outside the bond area. The LVDTs have been attached after the specimens were fixed into the testing machine. The strains along the CFRP strips were measured by attaching 6 strain gauges on their top faces, 5 within the overlap area and 1 outside. Three specimens (S-M514-30-1-I/II/III) were instrumented with 9 strain gauges in order to investigate the variation of the strains at smaller intervals. The instrumented configuration of the specimens is presented in Figure 5.

## 3. Experimental Results

All 34 specimens were loaded in tension up to failure. Based on the monitored parameters mentioned above, the following characteristics of the bonded joint were analyzed: failure modes, load–displacement behavior, strain distribution along the bond length at different loading stages, bond–slip relations and shear stress–strain variations of the CFRP strips.

### 3.1. Characteristic Failure Modes and the Corresponding Ultimate Loads

All failure mechanisms that were identified correspond to those already presented by other authors in previous research works [33,34,35,36,37,38,39]. However, only 6 of the 34 specimens failed by a single dominant mechanism, while the other 28 developed combined failure modes, characterized by two or three failure patterns. The failure mechanisms of some representative specimens are illustrated in Figure 6.

For the specimens bonded with the Sikadur 30 adhesive, the dominant failure mechanism was the cohesive one, while for those bonded with Sikadur 330 adhesive, the interface debonding mostly between the CFRP strip and the adhesive was the most common failure mode.

By analyzing the failure mechanisms of the specimens with identical configurations, it can be concluded that in most cases, when the predominant failure (cohesive or debonding at the composite–adhesive interface) is accompanied by secondary modes, such as the delamination of the CFRP strip or debonding at the steel–adhesive interface, the ultimate forces decrease considerably (especially for S-S512-330 specimens, where the ultimate forces are up to 55% lower).

For the specimens bonded with Sikadur 30 adhesive, the dominant failure mechanism was the cohesive one, which had a progressive behavior, starting from the loaded end and propagating gradually towards the opposite one. For the specimens made with Sikadur 330 adhesive, no cohesive failures were recorded. The most frequent failure modes were initiated at the CFRP strip–adhesive interface and occurred in a brittle and sudden manner.

The specimens with steel surfaces that were not sandblasted failed only by interface debonding between the steel plate and the adhesive at considerably lower ultimate loads when compared to the specimens having identical configurations but mechanically treated steel surfaces.

For the specimens having S512 CFRP strips, bonded with Sikadur 30 adhesive, it was observed that the ultimate forces were not directly influenced by the variations in the thickness of the adhesive layer. Thus, the highest values of the ultimate forces were obtained for the 1 mm adhesive thickness, and those recorded for thicknesses of 2 and 3 mm were smaller by 7% and 5%, respectively. For the same specimen configurations but a different type of adhesive (Sikadur 330), the lowest values of the ultimate forces were recorded for the bonded joints with 3 mm adhesive thickness.

Moreover, as it was expected, the failure mechanism strongly influenced the ultimate forces, as significant variations of the failure forces were recorded for specimens with identical configurations but a different failure pattern. The premature delamination of the CFRP strip caused an important decrease of the ultimate force: 37.5% for the 1 mm adhesive layer and 55% for the 3 mm adhesive layer. For the specimens with 2 mm adhesive thickness, the additional failure mode, developed at the steel–adhesive interface, led to an additional 13% decrease of the ultimate force, compared to the singular mode exhibited by the specimen that failed only at the CFRP–adhesive interface.

The same remarks are valid for the specimens made with M514 CFRP composite strip and bonded with Sikadur 30 adhesive. The changes in the thickness of the adhesive layer do not produce significant variations of the ultimate forces. However, for the M514 CFRP specimens bonded with Sikadur 330 adhesive, the maximum ultimate forces were recorded for the 1 mm adhesive layer, while the specimens having 2 and 3 mm adhesive thicknesses exhibited ultimate forces that were 38% and 47% smaller, respectively.

The effectiveness of the surface-treatment method was also demonstrated through the significant differences between the ultimate forces. All specimens that were not surface-treaded failed prematurely (predominantly at the steel–adhesive interface) at ultimate forces that were 33.65%–98.1% lower than the corresponding surface-treated configurations.

### 3.2. Load–Displacement Behaviour

The load–displacement distribution curves were plotted based on the values recorded during the testing process. Figure 7, Figure 8 and Figure 9 illustrate the load–displacement curves for some representative specimens.

As it can be observed in Figure 7a, for the S-S512-30 samples, two types of behavior characterized by different shapes and slopes of the load–displacement curves were obtained. In the first case, the curve displays an upward plateau followed by a second region of constant forces but rapidly increasing displacements. In the second case, the initial stiffness of the joint is smaller, and the slope of the curve does not undergo substantial changes during the loading stage. For specimens of the S-M514-30 type, the superior stiffness of the CFRP strip led to the diminishing of the ultimate displacements, which had maximum values of about 1.3 mm. The changes in the slope of the load–displacement curves can be correlated with the local failures that occurred along the bond length or with the transition from one failure mechanism to another.

For specimens of the S-S512-330 series (Figure 8a), the load–displacement curves are characterized by a single upward plateau with a slope that is significantly higher when compared to the one of the specimens of the S-S512-30 series. The ultimate displacements are in the range of 0.55–0.65 mm and do not vary significantly with respect to the thickness of the adhesive.

For the S-M514-330 specimens (Figure 8b), the load–displacement curves have similar shapes and slopes to those belonging to the S-S512-330 series. However, it can be observed that the maximum displacements decrease with respect to the stiffness of the CFRP strip, especially for the specimens bonded with the 1 mm adhesive layer, where the maximum displacements ranged between 0.3–0.35 mm.

For the specimens made with non-sandblasted steel plates (Figure 9), the load–displacement curves are similar to those of the corresponding surface-treated specimens. However, due to the premature interface debonding, the curves are characterized by only one ascending branch that stops at maximum displacements of 1 mm ended by the premature debonding. This proves that by sand-blasting the contact surface of the steel plates, the mechanical interlocking between the adhesive and the steel is significantly improved, especially due to the higher specific contact surface area.

### 3.3. Strain Distribution

The variation of the strains distribution along the bond length is one of the most important parameters in describing the specific characteristics of the bonding mechanism between steel elements and FRP composite strips [40]. During the experimental tests, the strains’ variations were recorded using the strain gauges fixed on the surface of the CFRP profiles. These values were later compared to the ones provided by the numerical analysis of the models. Figure 10, Figure 11 and Figure 12 show the strain distribution curves along the bond length at different load application rates for some representative specimens. The intervals between the fractions of the ultimate force at which the strain distribution curves were drawn are greater in the range 0.1–0.7 P_ult_ and smaller in the 0.85 P_ult_ - P_ult_ range in order to accurately identify the changes that occur before the failure of the specimens.

By comparing the strain distribution curves obtained for the specimens made with the Sikadur 30 adhesive to those of the specimens made with Sikadur 330, it is observed that in the first case, the values are much more scattered, thus demonstrating the progressive behavior of the cohesive failure mechanism. Furthermore, the ultimate values of the strains recorded for the specimens made with Sikadur 30 are around 4000 μm/m, while the ones recorded for the joints made with Sikadur 330 are greater, being close to 8000 μm/m for the S-S512-330-1 series.

It can be observed that with the increase in the axial stiffness of the composite strips, in the case of specimens made with M514 strips, the plateau of constant strain distributions extends more towards the free end of the composite elements, thus mobilizing a larger portion of the bond length. However, with respect to the thickness of the adhesive layer, no significant variations in the distribution and ultimate values of the strains were observed.

By comparing the strain distributions of the specimens made with non-sandblasted steel plates (NS-S512-330-1 and NS-M514-330-1 series) with the ones obtained for specimens with identical configuration, but with steel-treated surfaces (S-S512-30-1 and NS-M514-330-1 series), it can be concluded that for the first, the bond length is mobilized in a lower fraction, which is demonstrated by the narrow area on which the range of constant strain distribution extends.

### 3.4. Shear Stress–Displacement (Slip) Distributions

The shear stress–displacement curves, also referred to as bond–slip curves, establish a direct relationship between the values of the shear stresses and the corresponding displacements between the steel plate and the CFRP strip calculated at various key points located along the bond line [41]. In the case of SLJ specimens, the values of these two parameters can be obtained based on the strain variations recorded by the strain gauges by applying Equations (1) and (2) for the intervals between the positions of the strain gauges (Δ1–Δ5).

(1)$$\tau_{i/2}=\frac{\left(\varepsilon_{i}-\varepsilon_{i+1}\right)}{\left(L_{i+1}-L_{i}\right)}E_{CFRP}t_{CFRP}$$(2)$$\delta_{i/2}=\frac{\left(\varepsilon_{i}-\varepsilon_{i+1}\right)}{4}\left(L_{i+1}-L_{i}\right)+\sum_{i-1}^{n}\frac{\left(\varepsilon_{i+1}+\varepsilon_{i+2}\right)}{2}\left(L_{i+2}-L_{i+1}\right)$$
where:

*ε_i_* = the strain value recorded by the strain gauge “*i*”;

*n* = number of strain gauges; 

*L_i_* = the distance measured from the loaded end to the strain gauge “*i*”;

*E_CFRP_,**_FRP_* = elasticity modulus and width of the CFRP strip; 

*τ_i/2_* = the shear stress value computed at the middle distance between the strain gauge “*i*” and “*i*+1”;

*δ_i/2_* = the displacement value computed at the middle distance between the strain gauge “*i*” and “*i*+1”.

Figure 13, Figure 14 and Figure 15 show the shear stress–displacement (bond–slip) curves for some representative specimens. The main conclusion resulting from the analysis of the shear stress–displacement curves is that the shape and the slope of the curves are directly influenced by the adhesive characteristics and by the particularities of the dominant failure mode. In the case of specimens made with Sikadur 30 adhesive, the graph has a bi-linear pattern, characterized by an upward plateau and a descending one that is initiated immediately after reaching the ultimate shear stress. This behavior is common for the adhesive joints that fail in a cohesive manner since this specific failure mode is characterized by progressive development of stresses (mainly shear) up to the ultimate limit that corresponds to the shear failure of the adhesive.

For the specimens made with Sikadur 330 adhesive, in most of the cases, the shear stress–displacement curves are characterized by a single upward plateau. The failure mechanisms that were identified in the case of these joints occurred mostly at the adhesive–CFRP strip interface in a brittle and sudden manner (non-cohesive).

For the specimens made with non-sandblasted steel plates, the shear stress–displacement curves kept the same shape as those of the other specimens. For joints made with adhesive Sikadur 30, premature failure of the steel–adhesive interface led to a weaker mobilization of the bond length. The curves corresponding to the values recorded for the strain gauges no. 4–6 are suddenly interrupted, as can be observed in Figure 15b. The ultimate values of the shear stress for the specimens made with non-sandblasted steel plates are lower than of the corresponding specimens made with sandblasted steel plates, by approximately 28% for the S512 CFRP strip and 30% in the case M514 CFRP strip, respectively.

### 3.5. Stress–Strain Distributions for CFRP Strips

The strain variations in the free area of the CFRP strip were monitored during the loading stage using strain gauge no. 1 fixed to the upper side of the strip. The stresses were computed by dividing the load to the cross-sectional area of the CFRP strip, which has values of 60 mm^2^ for those of type S512 and 70 mm^2^ for those of type M514. Figure 16 and Figure 17 illustrate the stress–strain curves for some representative specimens.

By analyzing the stress–strain distributions at the level of the CFRP strips, it can be observed that, in general, the specimens that have the same configuration (i.e., CFRP strip, adhesive type and thickness), the stress–strain curves have similar shapes and ultimate values. For example, in the case of S512 CFRP strips, the maximum tensile stresses recorded in the free (unbounded) area of the CFRP are considerably smaller than the tensile strength of the adhesives (21% for specimens bonded with Sikadur 30 and 42% for Sikadur 330). The same remark is also valid for M514 CFRP strips, where the maximum tensile stresses were 23% and 44% of the tensile strength of Sikadur 30 and Sikadur 330 adhesives, respectively.

## 4. Numerical Modelling

### 4.1. Finite Element Model Description

The specimens were modeled using ANSYS Workbench software, the Static Structural module [42]. The 3D models were built as solid bodies, and the contact between the elements was considered as fully bonded using three interconnected primitives of parallelepiped shapes with eight nodes. Each node is characterized by three degrees of freedom. The parameters of connectivity and position were defined for each node to match the SLJ geometric configurations.

The adequacy of the finite element mesh was verified by the authors by progressively increasing the mesh density until the stresses, strain and displacements values converged by 1%. Thus, the final models were meshed using tetrahedral elements of 0.15 mm for the adhesive layer and hexahedral elements of 0.25 mm and 2.0 mm for the CFRP strip and steel plates, respectively. The size of the finite elements assigned for the steel plates was selected based on the results provided by the preliminary analysis, which had shown that no significant effects were produced at this level. A mesh refinement with the level set to 0.1 (0.1 mm maximum length of the mesh elements) was performed for the edges of the bond area since these regions are susceptible to stress concentrations [43,44]. The transition between the meshing areas with different elements and distinct refinement levels has been progressively achieved, with a reduced gradient, through specific zones, also called smooth transition regions. By modeling these regions, the risk of developing discontinuities in the mesh network is avoided. Additionally, by applying a smooth mesh and a fine refinement level, it is ensured that any variation in stresses and strains in the adhesive layer will be accurately determined [45]. Figure 18 presents the meshed models at the extremities of the bond length.

The metallic plates were modeled as S235 steel grade, with mechanical properties identical to those given in Eurocode 3 - EN 1993-1-1 (2006) [24]. The CFRP composite profiles were modeled as being linear-elastic, and orthotropic materials and their mechanical properties were applied based on the values presented in Table 2. The epoxy Sikadur 30 and Sikadur 330 adhesives were assumed to be linear-elastic, and isotropic materials and their physical and mechanical properties were taken from Table 3.

The boundary conditions were assessed in order to simulate the ones applied in the experimental program. Each model was considered to be fixed at the left bottom extremity, and the tensile load was applied at the opposite free end of the CFRP strip. The fracture criterion approach, valid for singular failure mechanisms, was not considered suitable to be applied in the numerical simulation since the failures that were recorded during the experimental program consisted of combined mechanisms and patterns at both the interface and adhesive levels. Therefore, the numerical models were loaded with the corresponding ultimate forces obtained during the experimental tests so as to determine the consistency between the experimental and numerical results with respect to the investigated parameters.

### 4.2. Results

The models were loaded in tension, and the variations in total displacement, shear stresses and strain along the bond area were monitored and compared to the experimental values. Figure 19 shows the total displacement maps for two representative specimens at the level of the CFRP strip.

The displacement variations obtained through the numerical analysis were compared to those recorded during the experimental testing. As can be observed from Figure 20, the ultimate values obtained through both analysis approaches are in good agreement. However, the numerical approach provides a linear load–displacement curve without displaying the changes in stiffness produced by local, intermediate debonding failures. Moreover, based on the experimental load–displacement curves, it can be observed that the specimens exhibited a local non-linear behavior contrarily to the linear elastic behavior provided by the manufacturer of the adhesives [20,21].

Based on the results of numerical modeling, the distributions of the shear stresses along the bond area were investigated. These distributions are presented in the form of chromatic maps, for two representative specimens (Figure 21), at the level of the adhesive layer.

By analyzing the shear stress variations based on the chromatic maps presented above, it can be seen that the maximum values developed near the loaded end, and their intensity gradually decreases towards the free end. Moreover, the ultimate values of the shear stresses, determined through experimental and numerical analysis approaches, are in good agreement.

Another parameter that was investigated based on the numerical analysis approach is the strain distributions along the contact area between the CFRP strips and the steel plates. These distributions are presented in the form of chromatic maps for two representative specimens (Figure 22) at the level of the CFRP strip. Additionally, Figure 23 presents the graphical representation of the strain distribution for both experimental and numerical approaches.

The comparative analysis of the strain distributions along the CPAF strip at different loading steps shows that for both the specimens made with adhesive Sikadur 30 and the specimens made with adhesive Sikadur 330, the ultimate strains calculated through numerical modelling have similar values to those obtained by experimental testing. Moreover, the shape and the slopes of the curves are similar in both cases. However, the local, intermediate failures that were recorded during the experimental tests were not precisely captured by the numerical simulation.

## 5. Discussions

Based on the results obtained by applying both experimental and numerical analysis approaches, the following aspects can be commented:The failure modes recorded during the experimental program correspond to the typologies already presented in previous research papers. The following failure typologies were identified: cohesive (shearing of the adhesive layer), CFRP strip delamination, steel –adhesive interface debonding and CFRP–adhesive interface debonding. However, only six specimens failed under a singular failure pattern, while the other 28 specimens failed under a combination of two or three failure mechanisms at both the interface and adhesive layers.By analyzing the variation of the ultimate forces, it was observed that for both S512/M514–30 and S512/M514–330 series, by increasing the thickness of the adhesive layer from 1 mm to 2 mm or 3 mm, the ultimate forces decrease up to 47% (the case of the S-M514-330-3 specimen).The load–displacement response is characterized by two distinct patterns. The first pattern, specific to the cohesive failures, consists of two distinct branches: one of constant variation between load and displacement, demonstrating the initial linear behavior, and the other, where the load is relatively constant, but the displacements are rapidly increasing. The second pattern, specific to failures at the interface levels, is characterized by a single branch of linear behavior interrupted by the sudden and brittle failure. This response was commonly specific to the specimens bonded with Sikadur 330 adhesive. The maximum displacements are directly influenced by the axial rigidity of the CPRF strip and by the thickness of the adhesive layer since the maximum displacements were recorded for the S512 strip and Sikadur 30 adhesive with 3 mm thickness.By analyzing the shear stresses–displacement (bond–slip curves) distributions, it can be observed that both for the joints made with S512 CFRP strips and for those made with M514 CFRP strips, the displacements corresponding to the ultimate shear stresses are varying with respect to the type and the thickness of the adhesive. Thus, in most of the cases, the ultimate values of the shear stress and displacements decrease with the increase in the adhesive layer thickness. Additionally, the ultimate values corresponding to the specimens bonded with Sikadur 330 adhesive are significantly higher than those of the specimens made with Sikadur 30 adhesive. However, the bond–slip curves can be used as a design tool only for the cohesive failures since the latter is controlled by the shear strength of the adhesive.By analyzing the strain variations along the bond line at different loading steps, it was observed that their distributions were initially concentrated near the loaded end. However, as the load increases, local debonding of the CFRP strip occurred and the strains distribution extended along the bond length. The limited extent of the strains demonstrates that the adhesive bonds are characterized by an effective bond length.Based on the numerical analysis of the finite element models, the chromatic maps were obtained in relation to the total displacement, shear stresses and shear strains. In addition, the strain distribution along the bond line at different loading steps was investigated. The numerical results are generally in good agreement with the experimental ones. Thus, the values of the ultimate displacements obtained by the two approaches are matched, both of them decreasing in relation to the axial rigidity of the CFRP strips. However, the non-linear behavior, characterized by the experimentally obtained distributions, cannot be simulated by numerical modelling since this approach involves a linear analysis. The shear stresses and strain distributions that were numerically computed confirmed that the bond length is loaded only within a limited region, validating the previous theories referring to the effective bond length. Thus, the maximum values develop near the loaded end of the CFRP strip, and their intensity gradually decreases towards the free end.Based on the results that were obtained through this combined experimental and numerical program, it can be concluded that the variation in the axial rigidity of the CFRP strip does not produce significant changes in the general behavior of the bonded system. On the other hand, the elastic and geometrical properties of the adhesive have a substantial impact on the structural response of the bond. The cohesive failures were obtained only for Sikadur 30 adhesive, characterized by a higher modulus of elasticity, while all specimens bonded with Sikadur 330, with a modulus of elasticity almost three times smaller than Sikadur 30, failed prematurely, by interface debonding or by delamination. This demonstrates that the shear strength of the adhesive is only partially used. Moreover, for SLJ bonded with adhesives having lower stiffness, the adhesion at the CFRP–adhesive interface should be further improved in order to avoid premature debonding. Generally, by increasing the thickness of the adhesive, the ultimate tensile forces decrease, probably due to the changes in the stress state, when the peeling stresses lead to premature debonding.

## 6. Conclusions

This paper presents the outcomes of a complex experimental and numerical study referring to the stress–strain behavior and failure particularities of single lap joints (SLJs) specimens composed of CFRP strips bonded to steel elements. 

By analyzing different geometrical and constitutive configurations of the bonded specimens, it has been observed that the failure mechanisms are of critical importance when assessing the efficiency of the joint. The most suitable one consists of cohesive failure since it mobilizes the largest percentage of the adhesive’s strength. However, the cohesive failures were obtained only for the adhesive with a high modulus of elasticity, while in the case of the other type of adhesive, having a modulus of elasticity almost three times smaller, all failures were premature by either debonding or delamination.

As was expected, the axial rigidity of the CFRP element does not produce important changes in the structural response of the bonded joint since it is mostly controlled by the adhesive’s characteristics and by the bonding effect. However, the adhesion mechanism is strongly influenced by the degree of preparing the contact surface. Thus, the sand-blasting method of surface treating the steel plates was efficient and produced a higher specific contact surface area.

The results of the numerical analysis were in generally good agreement with the experimental ones. The non-linear real behavior of the adhesives and the localized debonding effects that lead to premature failures cannot be accurately simulated by linear analysis. However, the distribution of the shear stresses and strains that were numerically computed confirmed that the bond length is loaded only within a limited region, thus validating the previous theories referring to the effective bond length.

## Figures and Tables

**Figure 1 materials-14-06722-f001:**
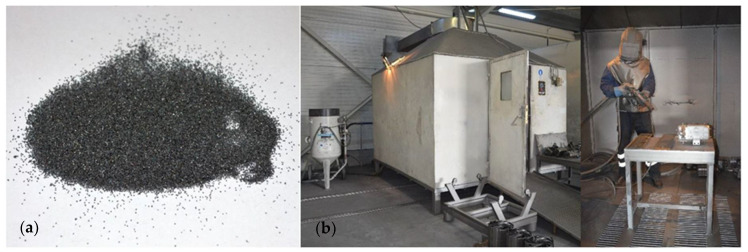
(**a**) GL-80 steel grits; (**b**) sandblasting chamber [28].

**Figure 2 materials-14-06722-f002:**
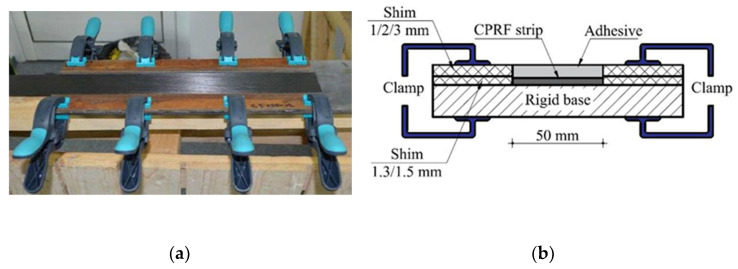
Rig for control of the adhesive thickness. (**a**) Laboratory setup, (**b**) Designed rig.

**Figure 3 materials-14-06722-f003:**
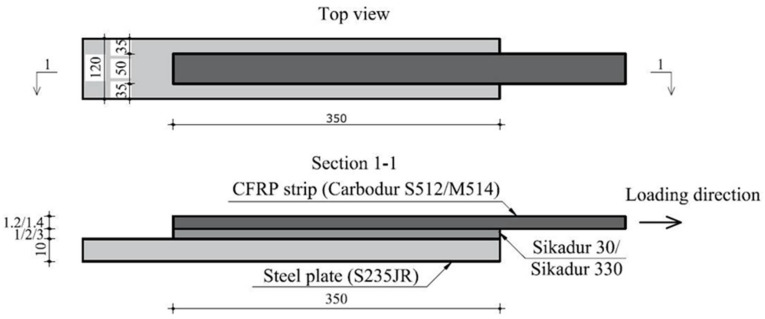
General layout of the specimens.

**Figure 4 materials-14-06722-f004:**
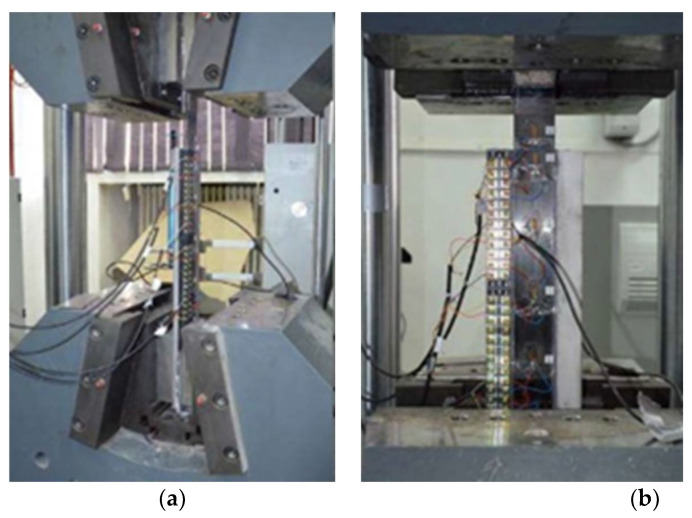
Test machine and specimen’s instrumentation. (**a**) Front view, (**b**) Lateral view.

**Figure 5 materials-14-06722-f005:**
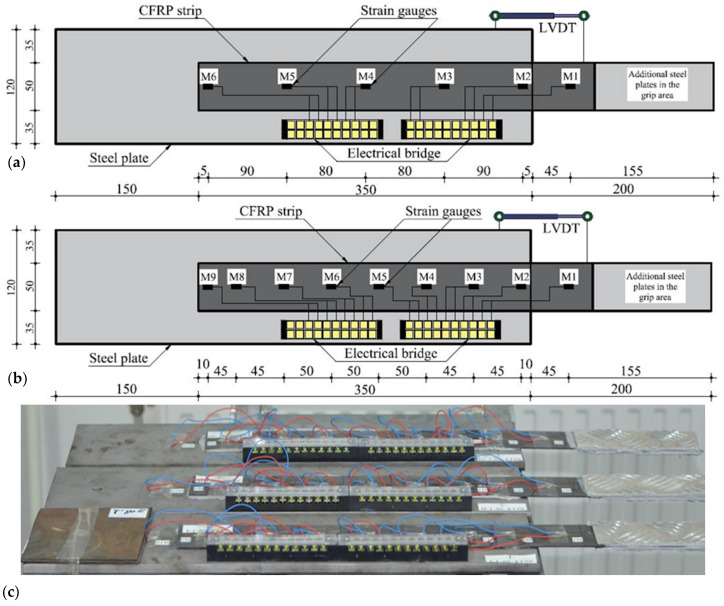
Specimens instrumentation with strain gauges and LVDT (dimensions in mm). (**a**) Graphical representation of specimens with 5 strain gauges, (**b**) Graphical representation of specimens with 9 strain gauges, (**c**) Real specimens with 9 strain gauges.

**Figure 6 materials-14-06722-f006:**
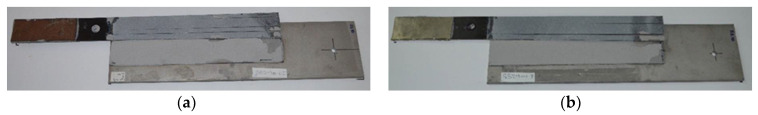
(**a**,**b**) Specimens 1 and 2, Failure mechanism: Cohesive adhesive layer failure (dominant) and CFRP strip delamination (secondary); (**c**,**d**) Specimens 8 and 10, Failure mechanism: CFRP–adhesive interface debonding; (**e**,**f**) Specimens 6 and 27, Failure mechanism: Cohesive adhesive layer failure (dominant), CFRP strip delamination (secondary) and steel–adhesive interface debonding (tertiary); (**g**,**h**) Specimens 33 and 34, Failure mechanism: steel–adhesive interface debonding (dominant) and CFRP–adhesive interface debonding (secondary).

**Figure 7 materials-14-06722-f007:**
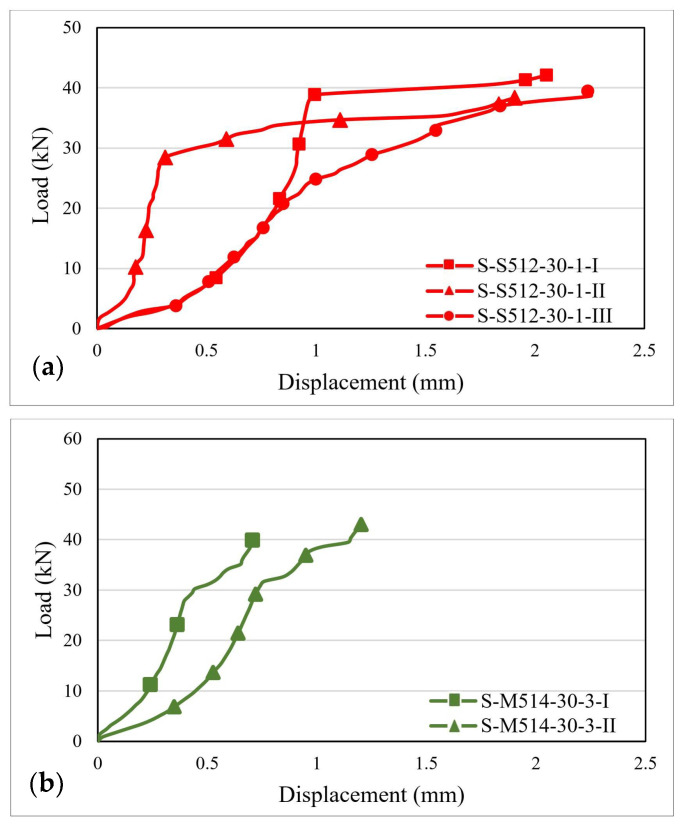
Load–displacement curves for: (**a**) S-S512-30-1 and (**b**) S-M514-30-3 specimens.

**Figure 8 materials-14-06722-f008:**
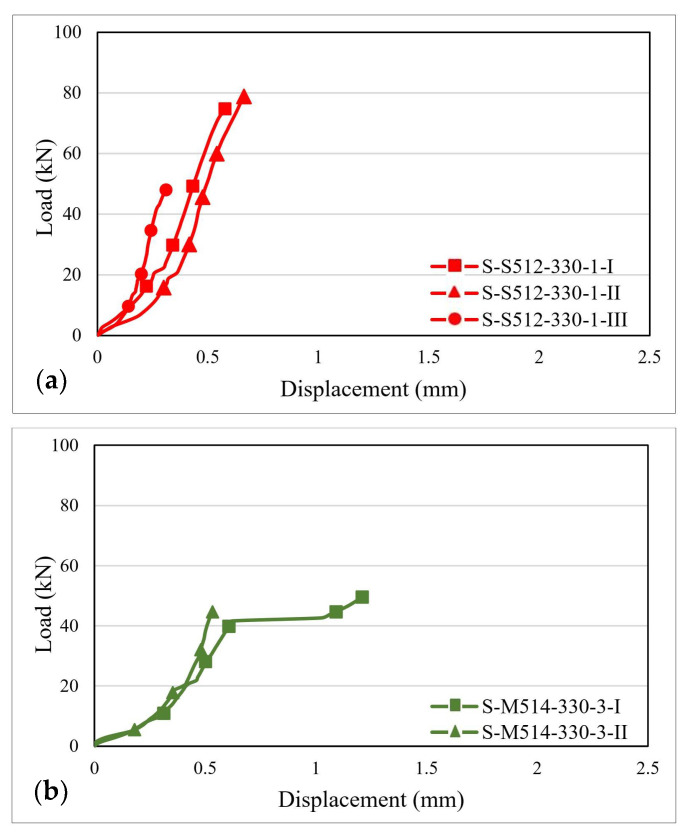
Load–displacement curves for: (**a**) S-S512-330-1 and (**b**) S-M514-330-3 specimens.

**Figure 9 materials-14-06722-f009:**
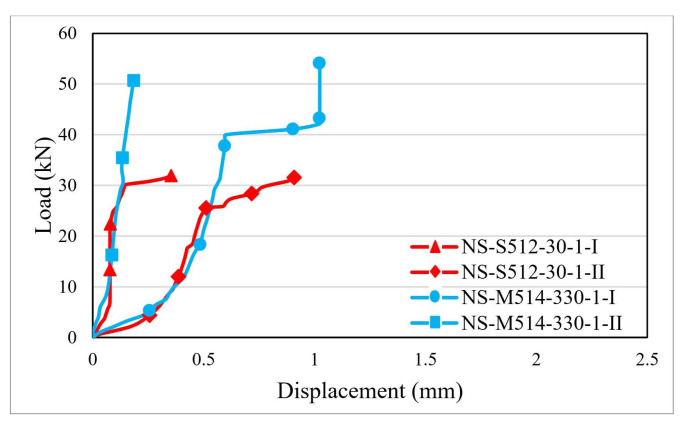
Load–displacement curves for specimens made with non-sandblasted steel plates.

**Figure 10 materials-14-06722-f010:**
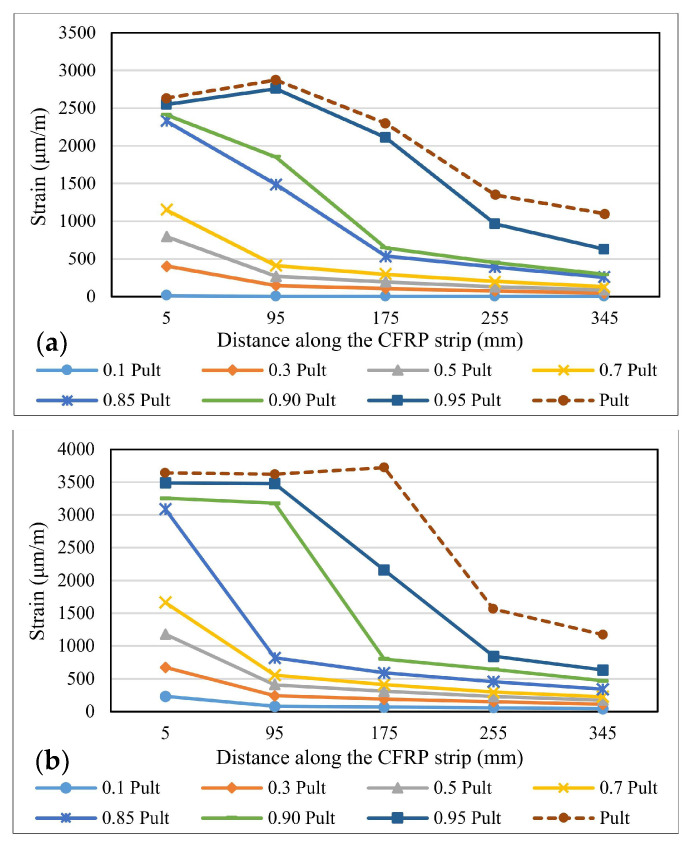
Strains distribution along the bond length for specimens: (**a**) S-S512-30-2-I and (**b**) S-M514-30-1-II.

**Figure 11 materials-14-06722-f011:**
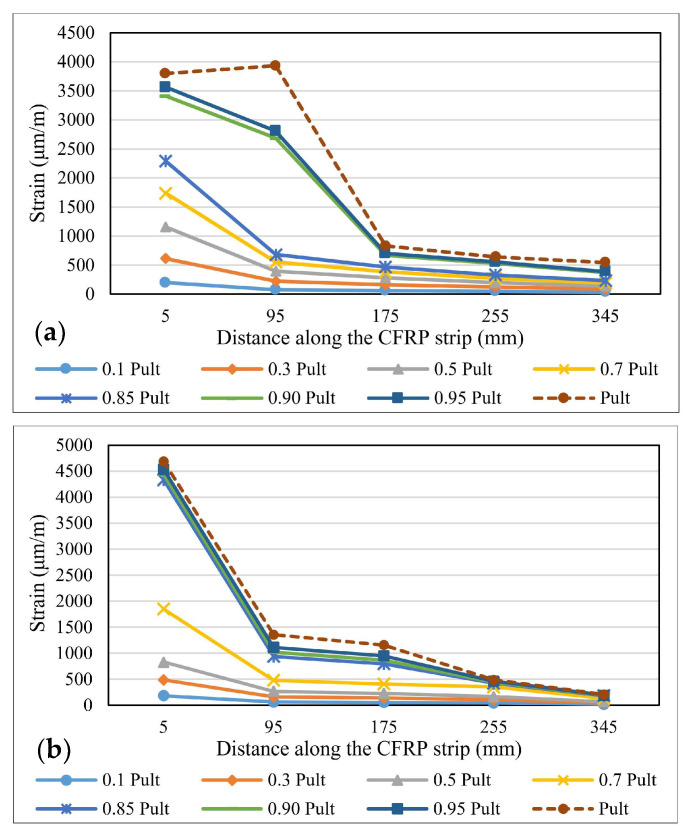
Strains distribution along the bond length for specimens: (**a**) S-S512-330-1-I and (**b**) S-M514-330-3-I.

**Figure 12 materials-14-06722-f012:**
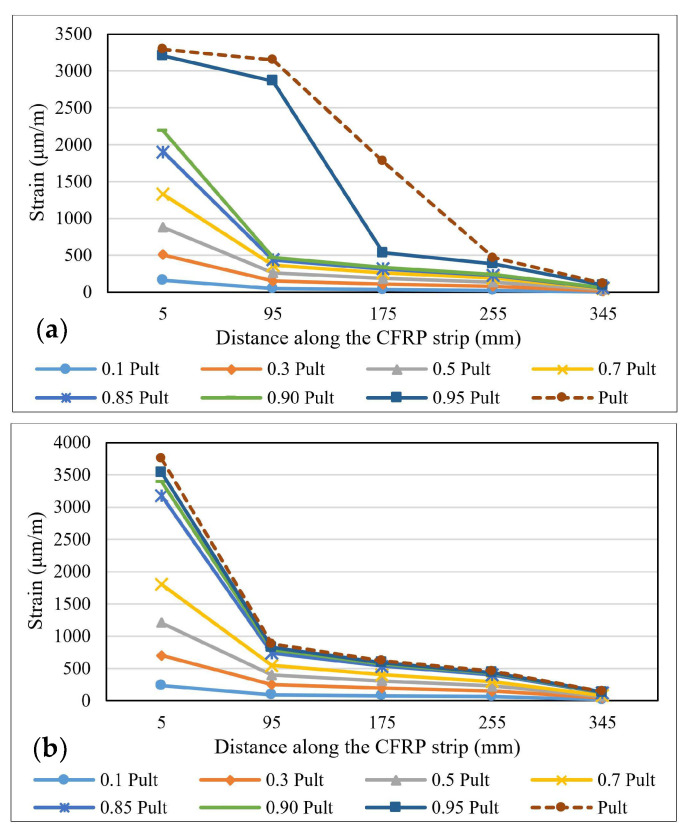
Strains distribution along the bond length for specimens: (**a**) NS-S512-30-1-I and (**b**) S-M514-330-3-I.

**Figure 13 materials-14-06722-f013:**
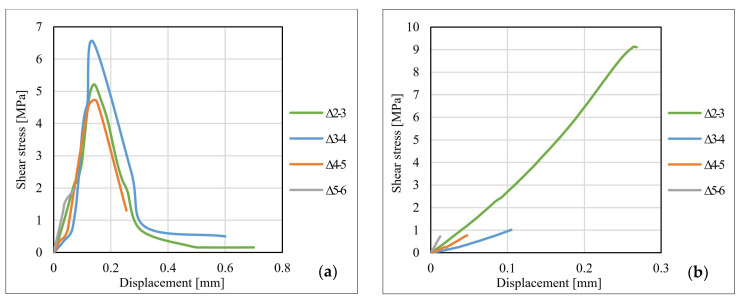
Bond–slip curves for specimens: (**a**) S-S512-30-2-I and (**b**) S-S512-330-1-II.

**Figure 14 materials-14-06722-f014:**
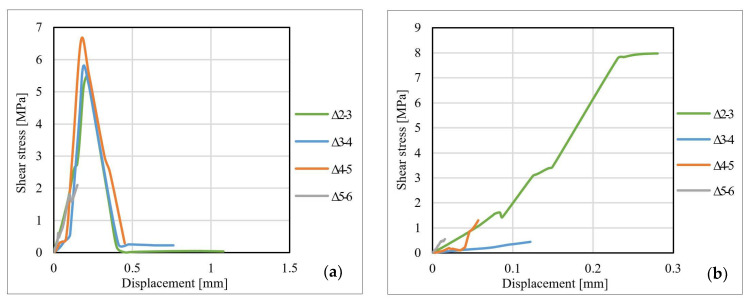
Bond–slip curves for specimens: (**a**) S-M514-30-3-I and (**b**) S-M514-330-3-I.

**Figure 15 materials-14-06722-f015:**
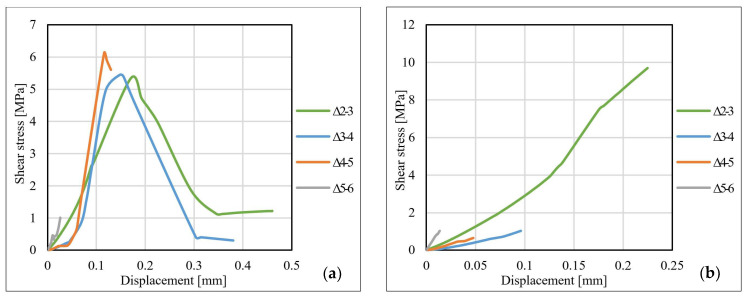
Bond–slip curves for specimens: (**a**) NS-S512-30-1-II and (**b**) NS-M514-330-1-I.

**Figure 16 materials-14-06722-f016:**
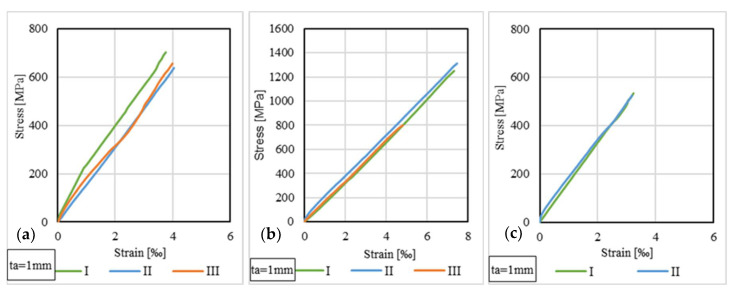
Stress–strain curves for S512 strip: (**a**) S-S512-30-1-I, II, III; (**b**) S-S512-330-1-I, II, III; (**c**) NS-S512-30-1-I, II.

**Figure 17 materials-14-06722-f017:**
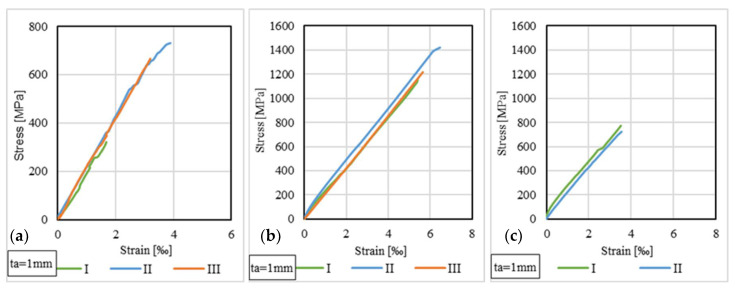
Stress–strain curves for M514 strip: (**a**) S-M514-30-1-I, II, III; (**b**) S-M514-330-1-I, II, III; (**c**) NS-M514-30-1-I, II.

**Figure 18 materials-14-06722-f018:**
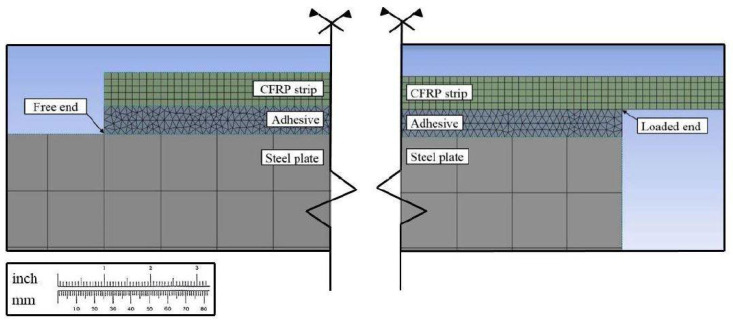
FE discretization at the extremities of the bond length.

**Figure 19 materials-14-06722-f019:**
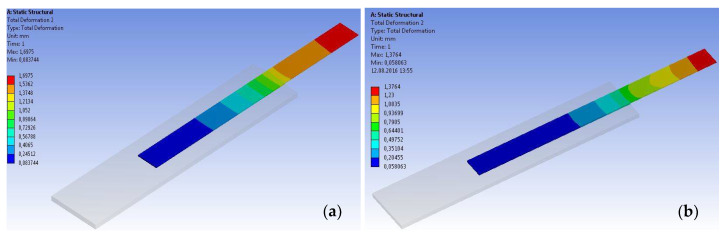
Displacement map at the level of the CFRP strip: (**a**) S-S512-30-1 (**b**) S-M514-30-2.

**Figure 20 materials-14-06722-f020:**
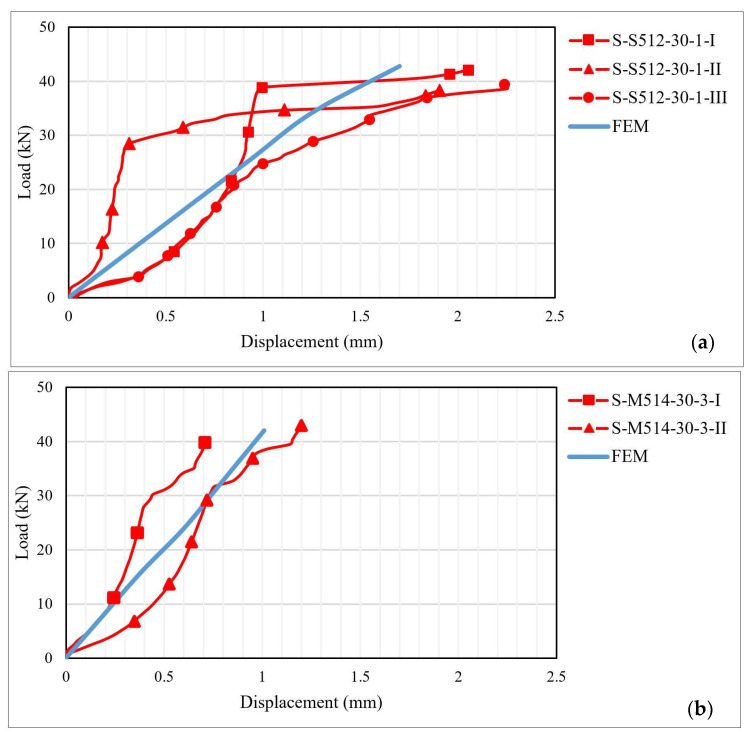
Load–displacement curves for specimens: (**a**) S-S512-30-1 and (**b**) S-M514-330-3 types.

**Figure 21 materials-14-06722-f021:**
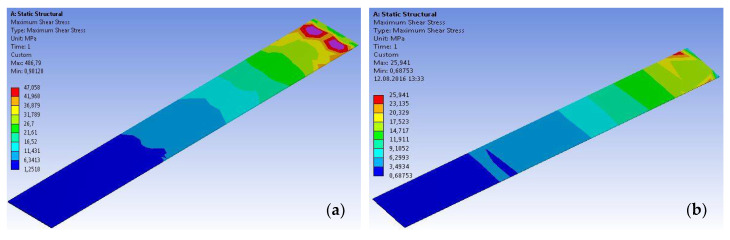
Typical shear stress distribution maps at the level of the adhesive layer: (**a**) S-S512-30-1 and (**b**) S-M514-30-1 types (only the adhesive layer is presented).

**Figure 22 materials-14-06722-f022:**
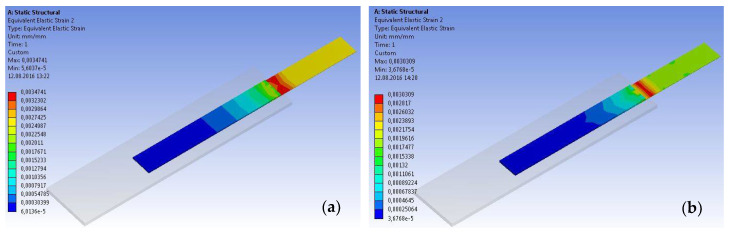
Typical strain distribution maps at the level of the CFRP strip: (**a**) S-S512-30-3 and (**b**) S-M514-30-3 types.

**Figure 23 materials-14-06722-f023:**
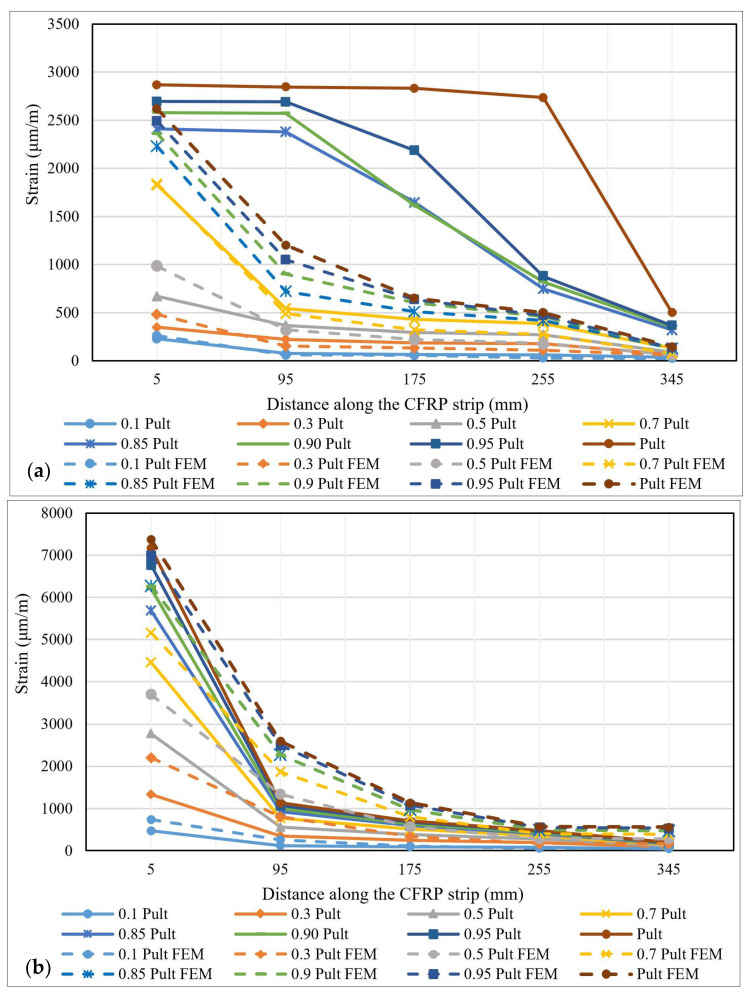
Strain distributions along the contact area between the CFRP strips and the steel plates for specimens: (**a**) S-M514-30-2 and (**b**) S-S512-330-1.

**Table 1 materials-14-06722-t001:** Properties of steel plates according to EN 1993-1-1:2006 [24].

Table S235.	S235 JR
Thickness, t_s_ (mm)	10
Width, b_s_ (mm)	120
Length, l_s_ (mm)	500
Yielding strength, f_y,s_ (N/mm^2^)	235
Ultimate strength in tension, f_u,s_ (N/mm^2^)	360
Modulus of elasticity, E_s_ (GPa)	210
Shear modulus of elasticity, G_s_ (GPa)	81
Poisson’s ratio, ν_s_	0.3
Thermal coefficient of expansion, α (1/^o^C)	12 × 10^−6^

**Table 2 materials-14-06722-t002:** Properties of the CFRP composite strips [18,19].

Properties	Sika Carbodur S512	Sika Carbodur M514
Density (kg/m^3^)	1600	
Fibre volume fraction (%)	≥68	
Thickness, t_CFRP,512/514_ (mm)	1.2	1.4
Width, b_CFRP_ (mm)	50	50
Length, l_CFRP_ (mm)	550	550
Cross sectional area (mm^2^)	60	70
Longitudinal modulus of elasticity, E_CFRP,512/514_ (GPa) (average values)	165	210
Tensile strength, f_t,CFRP,512/514_ (N/mm^2^)	3100	3200
Elongation at break, ε_u,CFRP,512/514_ (%) (minimum values)	>1.70	>1.35

**Table 3 materials-14-06722-t003:** Properties of the adhesives [20,21].

Type	Density (kg/m^3^) (Mixed)	Compressive Strength, f_c,a_ (N/mm^2^)	Tensile Strength, f_t,a_ (N/mm^2^)	Modulus of Elasticity, E_a_ (GPa)	Elongation at Break, ε_u,adh_ (%)
Sika 30	1650	70–80 (7 days, +10 °C)	25–28 (7 days, +15 °C)	12.8 (7 days, +15 °C)	1.0%
Sika 330	1300	30 (7 days, +23 °C)	33.8 (7 days, +23 °C)	4.5 (7 days, +23 °C)	0.9%

**Table 4 materials-14-06722-t004:** GL-80 steel grits produced by W Abrasive [28].

Grading	0.180 mm—99.8% 0.425 mm—0.2%
Hardness (Rockwell scale)	55.7 HRC
Density (kg/m^3^)	7600
Chemical composition	C ≥ 0.85%/P ≤ 0.05%/S ≤ 0.05%/0.40% < Si < 1.20%

**Table 5 materials-14-06722-t005:** Specimen details, ultimate loads and failure modes.

No.	Type	Adhesive Type	Adhesive Tensile Strength, f_t,a30/330_ (N/mm^2^)	Adhesive Modulus of Elasticity, E_a,30/330_ (GPa)	Adhesive Thickness, t_a,30/330_ (mm)	Ultimate Load, P_ult_ (kN)	Failure Mechanism
1	S-S512-30-1-I	Sikadur 30	25	12.80	1	42.10	C+D
2	S-S512-30-1-II	Sikadur 30	25	12.80	1	38.30	C+D
3	S-S512-30-1-III	Sikadur 30	25	12.80	1	39.40	C+D+SAI
4	S-S512-30-2-I	Sikadur 30	25	12.80	2	37.85	C+D+SAI
5	S-S512-30-2-II	Sikadur 30	25	12.80	2	38.65	C+D
6	S-S512-30-2-III	Sikadur 30	25	12.80	2	34.87	C+D+SAI
7	S-S512-330-1-I	Sikadur 330	30	4.50	1	74.80	CAI
8	S-S512-330-1-II	Sikadur 330	30	4.50	1	78.80	CAI
9	S-S512-330-1-III	Sikadur 330	30	4.50	1	48.00	CAI+D
10	S-S512-330-2-I	Sikadur 330	30	4.50	2	70.20	CAI
11	S-S512-330-2-II	Sikadur 330	30	4.50	2	61.25	CAI+SAI
12	S-M514-30-1-I	Sikadur 30	28	12.80	1	22.41	C+SAI
13	S-M514-30-1-II	Sikadur 30	28	12.80	1	51.06	C+SAI+D
14	S-M514-30-1-III	Sikadur 30	28	12.80	1	46.55	C+SAI+D
15	S-M514-30-2-I	Sikadur 30	28	12.80	2	10.00	C+D
16	S-M514-30-2-II	Sikadur 30	28	12.80	2	42.85	C+D
17	S-M514-330-1-I	Sikadur 330	30	4.50	1	80.80	SAI+CAI +D
18	S-M514-330-1-II	Sikadur 330	30	4.50	1	99.25	CAI
19	S-M514-330-1-III	Sikadur 330	30	4.50	1	85.00	CAI+SAI
20	S-M514-330-2-I	Sikadur 330	30	4.50	2	56.90	CAI+D
21	S-M514-330-2-II	Sikadur 330	30	4.50	2	53.35	CAI+SAI
22	S-S512-30-3-I	Sikadur 30	28	12.80	3	39.30	C+D+SAI
23	S-S512-30-3-II	Sikadur 30	28	12.80	3	38.75	C
24	S-S512-30-3-III	Sikadur 30	28	12.80	3	36.43	C+D
25	S-S512-330-3-I	Sikadur 330	30	4.50	3	23.10	CAI+D
26	S-S512-330-3-II	Sikadur 330	30	4.50	3	51.25	CAI
27	S-M514-30-3-I	Sikadur 30	28	12.80	3	39.80	C+D+SAI
28	S-M514-30-3-II	Sikadur 30	28	12.80	3	42.95	C+D
29	S-M514-330-3-I	Sikadur 330	30	4.50	3	49.65	CAI+D +SAI
30	S-M514-330-3-II	Sikadur 330	30	4.50	3	44.60	CAI+SAI
31	NS-S512-30-1-I	Sikadur 30	28	12.80	1	32.1	C+SAI
32	NS-S512-30-1-II	Sikadur 30	28	12.80	1	31.5	C+SAI
33	NS-M514-330-1-I	Sikadur 330	30	4.50	1	54.05	SAI+CAI
34	NS-M514-330-1-II	Sikadur 330	30	4.50	1	50.6	SAI+CAI

Failure mechanisms notations: C—Cohesive adhesive layer failure, D—CFRP strip delamination, SAI—Steel–adhesive interface debonding, CAI—CFRP–adhesive interface debonding.

## Data Availability

The data underlying this article will be shared on reasonable request from the corresponding author.

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
