# Peer review of "Structural Response of Bonded Joints between FRP Composite Strips and Steel Plates"

_materials, 2021, doi:10.3390/ma14216722_

Round 1
Reviewer 1 Report
The authors present the results of an experimental and numerical study on single lap joints (SLJs) between CFRP strips and steel elements loaded in tension up to failure. The study aims at assessing the effects on strength of parameters such as: composite material type, adhesive type and thickness.
It is this reviewer belief that the topic is of interest for the readers of the Journal, and that the results of the study are worthwhile for engineering application. The manuscript is appropriately organized and fairly well written. It is thus this reviewer opinion that it should be accepted to be published after minor revision following the recommendations here below.
At the end of the introduction the authors briefly state the aim of their work. They should consider conceiving a clearer, stronger, statement about the objectives of the research and about their contribution to the current state to the art.
Pag. 10-11, figs 7, 8, 9. The authors should consider using the same scale limits for the load. This will make the comparison of the different results easier for the readers.
Pag. 12-13, figs 10, 11, 12. The authors should consider using the same scale limits for the load at least for the graphs in the same figure.
Pag. 14 equation 1. Some of the symbols have a “GFRP” subscript that is wrong. CFRP is the correct one.
Pag. 14 Linea 330-336. Specifics relevant to equations 1 and 2 are difficult to read and contains typos (or extra equations that are not needed). Authors should rewrite this part eliminate unnecessary info and to make it clearer.
Pag. 15, figs 13, 14, 15. The authors should consider using the same scale limits for the load at least for the graphs in the same figure.
Pag. 14 lines 374-377. Sentences are not clear and prone to misunderstanding. Authors should consider rewriting.
Section 4.1. It seems that the authors developed a model of their specimen with enlarged dimensions. This is neither stated clearly nor justified in the text. Authors should consider adding missing info and justify their choices. Extra info shall be provided regarding loading and boundary conditions for the model.
Section 4. It is well established that epoxy adhesives exhibit a non-linear elasto-visco-plastic constitutive behaviour. The use and significance of nonlinear analyses should be justified and the relevance of the results that have been obtained adequately commented.
The conclusion section is wrongly numbered (should be section 5) and too long. The authors should consider moving the most part of the current contents to a discussion section and shorten the conclusions to a few paragraphs.
Author Response
The authors would like to thank the reviewer for the careful and thorough reading of this manuscript and for the thoughtful comments and constructive suggestions that help to improve the quality of this paper. We have incorporated all the suggestions made by the reviewer. Those changes are highlighted within the manuscript. Please see below a point-by-point response to the reviewers’ comments and concerns.
At the end of the introduction the authors briefly state the aim of their work. They should consider conceiving a clearer, stronger, statement about the objectives of the research and about their contribution to the state of the art.
- A new paragraph has been added at the end of chapter 1 (lines 67-71 highlighted in yellow)
Pag. 10-11, figs. 7, 8, 9. The authors should consider using the same scale limits for the load.
Pag. 12-13, figs. 10, 11, 12. The authors should consider using the same scale limits.
- The limits of the scales were set in such a way that the general response of the specimens (with respect to different parameters) is presented in the most illustrative way and also by also taking into account the size limitations which are imposed by the journal’s format. We are kindly recommending to present the figures with the scales provided by the authors in order to better understand the behavior of the bonded joints.
Pag. 14, equation 1. Some of the symbols have GFRP subscript that is wrong.
- The error has been corrected (highlighted in yellow).
Pag. 14 Linea 330-336. Specifics relevant to equations 1 and 2 are difficult to read and contains typos (or extra equations that are not needed). Authors should rewrite this part eliminate unnecessary info and to make it clearer.
- The description of the equation’s terms has been rewritten (lines 339 – 346, highlighted in yellow)
Pag. 15, figs 13, 14, 15. The authors should consider using the same scale limits for the load at least for the graphs in the same figure.
- The limits of the scales were set in such a way that the general response of the specimens (with respect to different parameters) is presented in the most illustrative way and also by also taking into account the size limitations which are imposed by the journal’s format. We are kindly recommending to present the figures with the scales provided by the authors in order to better understand the behavior of the bonded joints.
Pag. 14 lines 374-377. Sentences are not clear and prone to misunderstanding. Authors should consider rewriting.
- The entire paragraph after Figure 17 has been reformulated in order to make it clearer (lines 383-391 highlighted in yellow).
Section 4.1. It seems that the authors developed a model of their specimen with enlarged dimensions. This is neither stated clearly nor justified in the text. Authors should consider adding missing info and justify their choices. Extra info shall be provided regarding loading and boundary conditions for the model.
- The 3D model in ANSYS had the exact same dimensions and characteristics as the experimental samples. The misunderstanding came from an error in presenting the sizes and types of the FEM discretization elements. (corrected above as required at no. 3). The boundary conditions and loading method are described in section 4.1. (lines 419-434).
Section 4. It is well established that epoxy adhesives exhibit a non-linear elasto-visco-plastic constitutive behaviour. The use and significance of nonlinear analyses should be justified and the relevance of the results that have been obtained adequately commented.
The results of the numerical modelling are presented in section 4.2. and also in section (5).
- The linear elastic analysis has been selected as suitable in order to describe the general response of the specimens. Also, because of the very different failure mechanisms that were recorded during the experimental program (complex and multiple failure patterns), the authors did not found any relevant failure criteria to be applied in the numerical analysis in order to be similar to the experimental results.
The conclusion section is wrongly numbered (should be section 5) and too long. The authors should consider moving the most part of the current contents to a discussion section and shorten the conclusions to a few paragraphs.
- A new section has been added (no. 5) where the outcomes of the experimental program and the results of numerical investigations are presented and discussed. The conclusion were rewritten and presented only briefly in a separate section (no. 6).

Reviewer 2 Report
This work reports on an experimental and modelling study of the tensile behavior of various types of epoxy bonded steel and CFRP. The work will serve to be useful for material scientists working on adhesives and composites. It can be accepted provided that the following comments are sufficiently addressed.
-Line 250, spelling error. Should be surface-treated. Define FEM when the term first appears in the abstract.
-Can the authors at least give some insights on why the sand-blasted steel has got stronger bonding with the CFRP? The reviewer presumes that it got to do with the more “biting” points, and higher specific surface area.
-The authors should mention clearly in the text that the Static Structural solver in Ansys Workbench was used. The authors should also specify that solid bodies were modelled, as opposed to shells (if the reviewer did not interpret it wrongly). For the contact between the three bodies, are they set to be fully bonded? The authors should specify.
-Line 387. If the simulation is done using solid bodies, I think the authors meant tetrahedral and hexahedral meshes? As opposed to triangular, rectangular and square, which are improper terms.
-The authors should give more insights on the difference between the FEM and experimental results (e.g. Figure 20). The reviewer think that this may owe to some localized debonding, improper tensile grip etc. The epoxy may also be inherently nonlinear despite the manufacturer data. The authors are can just give some comments based on these in the text.
-For Figure 19, 21 and 22, the authors need to specify clearly what the body shown (with color distribution) is, like is it the epoxy adhesive itself, either in the text or labels the figures.
Author Response
The authors would like to thank the reviewer for the careful and thorough reading of this manuscript and for the thoughtful comments and constructive suggestions that help to improve the quality of this paper. We have incorporated all the suggestions made by the reviewer. Those changes are highlighted within the manuscript. Please see below a point-by-point response to the reviewers’ comments and concerns. All page numbers refer to the revised manuscript file with tracked changes.
-Line 250, spelling error. Should be surface-treated. Define FEM when the term first appears in the abstract.
- The spelling error has been corrected (highlighted in yellow). FEM was defined in the abstract (highlighted in yellow)
-Can the authors at least give some insights on why the sand-blasted steel has got stronger bonding with the CFRP? The reviewer presumes that it got to do with the more “biting” points, and higher specific surface area.
- The aspect has been additionally commented in the first paragraph after Figure 9 (lines 292-294, highlighted in yellow)
-The authors should mention clearly in the text that the Static Structural solver in Ansys Workbench was used. The authors should also specify that solid bodies were modelled, as opposed to shells (if the reviewer did not interpret it wrongly). For the contact between the three bodies, are they set to be fully bonded? The authors should specify.
- The required information has been added to the first two paragraphs from section 4.1. (lines 393-395, highlighted in yellow)
-Line 387. If the simulation is done using solid bodies, I think the authors meant tetrahedral and hexahedral meshes? As opposed to triangular, rectangular and square, which are improper terms
- The changes were made according to the review’s requirements in the second paragraph from section 4.1 (lines 402-403).
-The authors should give more insights on the difference between the FEM and experimental results (e.g. Figure 20). The reviewer think that this may owe to some localized debonding, improper tensile grip etc. The epoxy may also be inherently nonlinear despite the manufacturer data. The authors are can just give some comments based on these in the text.
- The authors presented the non-linear experimentally obtained behavior (see lines 530-533). However, additional comments have been added in the second paragraph from section 4.2. (lines 446-448, highlighted in yellow).
-For Figure 19, 21 and 22, the authors need to specify clearly what the body shown (with color distribution) is, like is it the epoxy adhesive itself, either in the text or labels the figures.
- For each of the three figures (19, 21 and 22) additional comments have been provided in the paragraphs where the figures are presented (lines 438, 453, 466) and in the titles of the figures (lines 440, 456, 470) (highlighted in yellow.
